# LOGO-VGR: VISUAL GROUNDED REASONING FOR OPEN-WORLD LOGO RECOGNITION

## ABSTRACT

Recent advances in multimodal large language models (MLLMs) have been primarily evaluated on general-purpose benchmarks, while their applications in domain-specific scenarios, such as intelligent product moderation, remain underexplored. To address this gap, we introduce an open-world logo recognition benchmark, a core challenge in product moderation. Unlike traditional logo recognition methods that rely on memorizing representations of tens of thousands of brands—an impractical approach in real-world settings—our proposed method, Logo-VGR, enables generalization to large-scale brand recognition with supervision from only a small subset of brands. Specifically, we reformulate logo recognition as a comparison-based task, requiring the model to match product images with candidate logos rather than directly generating brand labels. We further observe that existing models tend to overfit by memorizing brand distributions instead of learning robust multimodal reasoning, which results in poor performance on unseen brands. To overcome this limitation, Logo-VGR introduces a new paradigm of domain-specific multimodal reasoning: Logo Perception Grounding injects domain knowledge, and Logo-Guided Visual Grounded Reasoning enhances the model's reasoning capability. Experimental results show that Logo-VGR outperforms strong baselines by nearly 10 points in OOD settings, demonstrating superior generalization.

## 1 INTRODUCTION

In recent years, multimodal large language models (MLLMs)(Bai et al., 2025; Wu et al., 2024; Achiam et al., 2023) have been widely applied across various scenarios. Beyond direct zero-shot problem solving, many studies have emphasized post-training paradigms, such as SFT and PPO(Schulman et al., 2017), as well as GRPO(Shao et al., 2024), to adapt models to downstream tasks better. Several open-source MLLMs, such as Qwen2.5-VL(Bai et al., 2025), have significantly accelerated the deployment of intelligent applications.

To evaluate the performance of multimodal large language models (MLLMs) and post-training methods in domain-specific applications, we propose a benchmark for real-world scenarios: open-world logo recognition, which is a crucial task in product moderation. In traditional logo recognition methods, models are trained to learn and optimize representations for each class and then output classification results. This approach often leads to overfitting and memorization of class-specific features. However, with tens of thousands of brands in the open world, it is impractical to train models to recognize such an enormous number of classes while still ensuring high accuracy. Rather than relying on memorization, humans recognize unseen brands by comparing product images with candidate logos—a reasoning process that naturally generalizes to these unseen brands.

Motivated by this, we reformulate the task by framing logo recognition as a comparison task rather than directly predicting brand labels, requiring the model to match product images with candidate logos. Ideally, this comparison-based formulation could lead to better generalization. To explicitly evaluate the model's ability to generalize to unseen brands, our benchmark is divided into in-domain (ID) and out-of-domain (OOD) test sets, with OOD brands being entirely absent from the training data. During evaluation, candidate logos are provided as references, and the model is required to determine the correct match by comparing the product image with reference logos.

As illustrated in Fig. 1, we observe that directly applying SFT or GRPO for answer supervision improves accuracy on ID data but degrades generalization to OOD data. This suggests that the model tends to memorize answer-specific knowledge while overlooking the development of general reasoning capabilities. To address this limitation, we propose Logo-VGR, a domain-specific multimodal reasoning paradigm that learns generalized reasoning from a small amount of logo recognition data and achieves strong generalization performance on OOD data.

To mitigate the shortcut learning phenomenon where models rely on memorization during training, we employ GRPO with carefully designed rewards to guide the model toward more generalized reasoning. Specifically, motivated by GRIT (Fan et al., 2025), we encourage the model to explicitly output coordinate-based evidence during reasoning—i.e., predicting the location of logos in the image for the logo recognition task. To ensure that the model genuinely solves the task, we supervise these visual Coordinate Clues using an IoU-based metric. In particular, inspired by Vision-R1 (Zhan et al., 2025), we calculate precision and recall for the predicted coordinates and employ them as reward signals. In addition, to regulate the model's Cognitive Trajectory, we leverage a large language model as a judge to evaluate the quality of its reasoning process.

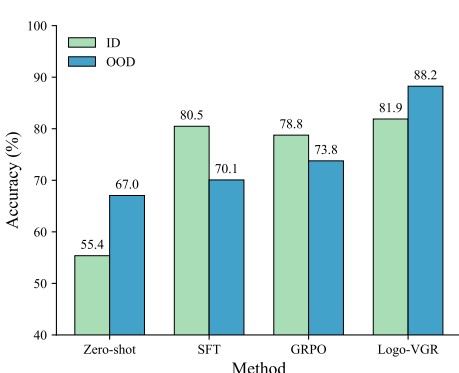

Figure 1: The accuracy results of different methods on ID and OOD benchmarks. Here, zero-shot refers to the Qwen2.5-VL-3B baseline. Through SFT training, the model improves its performance on ID data but simultaneously suffers from reduced generalization ability. In contrast, Logo-VGR leverages process supervision to encourage correct reasoning, thereby achieving stronger generalization.

Another critical challenge is equipping the model with fundamental domain knowledge before addressing downstream tasks. In the context of logo recognition, pretrained models generally focus on generic objects and therefore lack sufficient logo-specific perception. To address this, we introduce a proxy logo detection task, requiring the model to output the absolute coordinates of logos in JSON format. Our training follows a two-stage paradigm: (1) Supervised Fine-Tuning for Domain Knowledge Transfer, which explicitly teaches the model domain-specific knowledge, and (2) Spatially-Aware Reward Design, which further enhances the model's spatial awareness through reinforcement learning.

Overall, our contributions can be summarized as follows:

- We construct a real-world multimodal benchmark for open-world logo recognition. With in-domain (ID) and out-of-domain (OOD) splits, we focus on evaluating MLLMs' domain-specific reasoning and their generalization to unseen brands.

- We propose Logo-VGR, a novel post-training paradigm in domain-specific scenarios, which strengthens reasoning generalization through a two-stage training strategy.

- Experimental results demonstrate that Logo-VGR achieves significant improvements on both ID and OOD test sets, with nearly 10-point gains on OOD tasks.

## 2 LOGO RECOGNITION BENCHMARK

### 2.1 TASK DESIGN

The simplified illustration of the Logo Recognition Benchmark is shown in Fig. 2. The task requires the model to identify the brand associated with a given product image. To prevent the model from simply memorizing brand distributions, we reformulate the generative problem into a comparison task: the model is required to select the correct brand from three candidate brands coarsely retrieved by a detection model. To evaluate generalization ability, we split the test set into ID and OOD subsets. We aim for the model to learn generalizable reasoning skills on the ID training set, which can then be transferred to unseen OOD brands. We find that relying solely on direct supervision

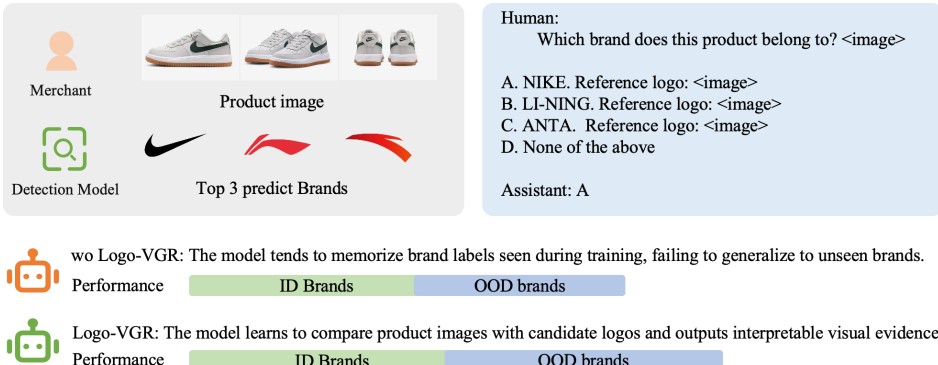

Figure 2: Overview of the Logo Recognition Benchmark. To prevent the model from memorizing brand information, we reformulate the original memory-based task into a comparison-based one, where the model is required to compare the features of product images with those of candidate logos to produce the answer.

with the answers encourages the model to memorize brands and achieve better performance on the training set, but it consequently results in poor generalization to OOD brands. In contrast, our Logo-VGR method explicitly teaches the model how to reason during training, substantially enhancing its ability to generalize.

**Why choose Logo Recognition as the task?** Existing studies primarily evaluate models on general-domain datasets. However, the broader application of large models lies in diverse downstream tasks, and a key challenge is how to leverage post-training to achieve strong performance in specialized domains. This challenge is particularly critical for multimodal large models, which have great potential economic value in replacing labor-intensive processes such as intelligent content auditing. We therefore choose Logo Recognition, a core task in product auditing on e-commerce platforms, to build our benchmark, which aims to evaluate how post-training can effectively adapt models to real-world domain-specific problems.

**Why reformulate logo generation into a comparison-based task?** In traditional classification settings, the model is required to memorize the features of each brand for recognition. This approach is infeasible for millions of brands and does not generalize to unseen brands. To address this, we reformulate the original generative task into a comparison-based task: the model is given three candidate brand logos retrieved by a coarse detection model and is required to select the correct one. For rare cases where the correct brand is not recalled, we add a "None of the above" option. The coarse retrieval model consists of a logo detector and a representation–retrieval module. The logo detector localizes potential logos in the product image, crops them for representation learning, and then performs retrieval against a million-scale logo database.

**Why split the test set into ID and OOD subsets?** According to the (World Intellectual Property Organization, 2024), as of 2023, there were approximately 88.2 million active trademark registrations worldwide, and platforms such as TikTok host more than 5 million brands(London, 2023). Clearly, brand recognition by memorizing every brand is infeasible. Instead, the model must be capable of generalizing by learning transferable multimodal reasoning strategies during training. To evaluate this, we partition the test set based on brand identity: ID represents brands seen dur-

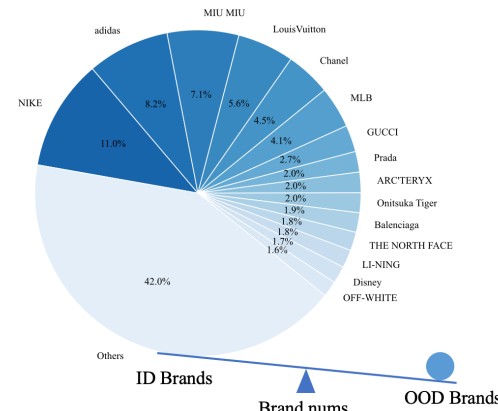

Figure 3: Statistics of brand distribution. A small number of top brands account for the majority of occurrences. We categorize the top brands as ID brands and the remaining brands as OOD brands.

ing training, while OOD represents brands absent from the training set. Only by acquiring general multimodal reasoning abilities can a model perform well across both ID and OOD settings.

## 2.2 DATA STATISTICS

The training set contains 7,409 question–answer pairs spanning 373 brands across 107 categories. The test set consists of 1,853 question–answer pairs, while the OOD test set contains 1,814 question–answer pairs involving 109 entirely novel brands. Detailed brand distribution statistics can be found in the appendix. To facilitate coordinate prediction for multiple images, we concatenate several product images (on average, five per instance) and adjust their coordinates accordingly. The image resolution ranges from $224 \times 224$ to $1024 \times 1024$.

In real-world scenarios, brand distributions exhibit a long-tail pattern, where a small number of head brands account for the vast majority of instances. We categorize the head brands as the training set and randomly sample the tail brands to construct an OOD test set for evaluating the model's generalization ability to unseen brands. The brand distribution of the dataset is illustrated in Fig. 3. Consequently, due to the inherent distribution of real-world data, the model must generalize effectively to numerous OOD brands after being trained on only a limited set of head brands.

## 3 RELATED WORK

**Multimodal Large Language Models.** Large language models (LLMs) have made remarkable strides in text-based applications; however, their capabilities remain limited when faced with the growing prevalence of vision-centric tasks. To bridge this gap, recent research has extended LLMs into the visual modality, giving rise to multimodal large language models (MLLMs). Representative efforts such as LLaVA Liu et al. (2023), LLaMA (Touvron et al., 2023), Qwen-VL (Bai et al., 2023), InternVL (Chen et al., 2024), and DeepSeek-VL (Lu et al., 2024) leverage large-scale vision-language pretraining to strengthen perceptual grounding and cross-modal reasoning. In addition to large-scale pretraining, post-training techniques play a crucial role in adapting MLLMs to specific downstream benchmarks and real-world scenarios. Instruction tuning and task-oriented fine-tuning have been shown to improve alignment with human supervision, while reinforcement learning approaches (Shao et al., 2024; Rafailov et al., 2023) further refine model reasoning and output faithfulness. These strategies have demonstrated effectiveness across specialized domains, including object detection (Zhan et al., 2024a) and semantic segmentation (Wei et al., 2024; Yang et al., 2023), highlighting the versatility and growing potential of multimodal alignment techniques.

**Visual Grounded Reasoning.** Significant progress has been achieved in text-based Long-CoT(Guo et al., 2025), and many studies have investigated how to extend model reasoning to the multimodal domain. Modal-bridging approaches(Huang et al., 2025; Yang et al., 2025) generate image descriptions with multimodal models and then feed those descriptions into Long-CoT language reasoning models to produce reasoning traces, thereby enabling an indirect treatment of vision–language tasks. To strengthen multimodal reasoning models' attention to image content, "thinking with images" methods(Fan et al., 2025; Wang et al., 2025a; Jiang et al., 2025) guide models to output explicit spatial coordinates during reasoning so as to better ground the inference in the visual input. Other approaches steer models to use tools(Zheng et al., 2025b; Wang et al., 2025b) for image processing, thereby generating multimodal chains of thought.

**Logo Recognition.** Previous works have introduced numerous logo detection datasets, such as OpenBrand (Jin et al., 2020), LogoDet-3K (Wang et al., 2022), and SalECI (Jiang et al., 2022), where traditional computer vision-based detection methods (Yuan et al., 2025; Jia et al., 2024) have achieved significant progress in this domain. In the context of logo classification, prior approaches (Hou et al., 2024) leveraged contrastive learning between CLIP text embeddings and logo image representations for classification. However, under this paradigm, models tend to rely heavily on optimizing and memorizing category distributions, which constrains their capacity for true multimodal reasoning and impedes generalization. In this work, we focus on addressing product recognition in real-world complex scenarios using MLLMs, aiming to enhance their capability for analogical reasoning by guiding the models toward correct reasoning processes. Unlike prior methods, our approach emphasizes reasoning-oriented supervision rather than category memorization, thereby enabling MLLMs to achieve stronger generalization and robustness in open-world settings.

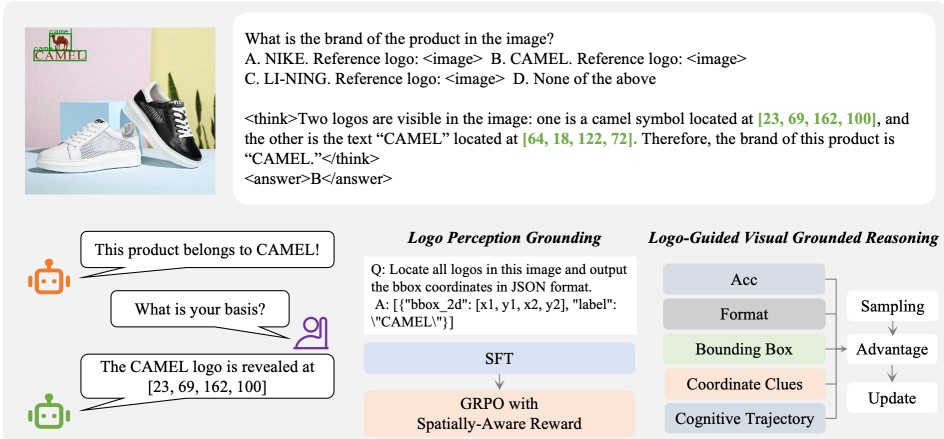

Figure 4: Illustration of the Logo-VGR framework. Logo-VGR is a two-stage, domain-adaptive multimodal reasoning pipeline. In the first stage, domain knowledge is enhanced via logo detection to improve low-level logo perception. In the second stage, Logo-Guided Visual Grounded Reasoning is introduced to prevent shortcut reasoning based on logo-style memorization and to guide the model toward a more principled and generalizable multimodal reasoning paradigm.

# 4 METHOD

## 4.1 OVERVIEW

To enhance the model's generalization capability on domain-specific tasks, we propose Logo-VGR, a two-stage, domain-adaptive multimodal reasoning pipeline, as seen in Fig. 4. First, we enhance the model's domain knowledge by leveraging logo detection tasks to improve its low-level logo perception. To avoid shortcut reasoning based on logo-style memorization, we introduce Logo-Guided Visual Grounded Reasoning, directing the model toward a principled and generalizable multimodal reasoning paradigm.

- **Stage 1: Logo Perception Grounding.** An auxiliary detection task is employed to strengthen the model's ability to perceive logos in domain-specific scenarios. Coupled with a two-step SFT + GRPO paradigm, this stage effectively enhances both logo perception and brand awareness.
- **Stage 2: Logo-Guided Visual Grounded Reasoning.** The model's reasoning process is further refined via reinforcement learning to develop a more generalizable inference strategy. To avoid shortcut reasoning driven by memorized logo patterns, Logo-VGR explicitly supervises both the reasoning trajectory and the intermediate bounding-box coordinate predictions.

## 4.2 LOGO PERCEPTION GROUNDING

To strengthen the model's domain-specific understanding, we perform continual pretraining within the target domain. For logo recognition, we introduce an auxiliary detection task that enhances the model's capability in perceiving logos.

Most existing MLLM-based detection approaches (Zhan et al., 2024a;b) perform direct supervised fine-tuning, where the model is trained to predict bounding boxes formatted as JSON. Recent works(Zhan et al., 2025) have shown that reinforcement learning with IoU-based rewards achieves superior performance and generalization compared with pure SFT. However, these methods often assume that the model has been exposed to similar data during large-scale pretraining, which facilitates refinement through sampling-based optimization.

For domain-specific scenarios, we propose a two-step strategy: SFT is first employed to inject new knowledge explicitly, followed by GRPO (Shao et al., 2024) to further optimize the model using spatially-aware rewards.

**Supervised Fine-Tuning for Domain Knowledge Transfer.** For unseen knowledge, token-level cross-entropy supervision enables the model to quickly acquire new concepts. However, conventional cross-entropy fails to consider spatial relationships. For example, coordinates "100" and "99" are spatially adjacent, yet cross-entropy treats them as entirely different tokens, leading to overfitting on the discrete training distribution while ignoring coordinate continuity. To mitigate this, we introduce IoU-based reward signals to reinforce spatial awareness.

**Spatially-Aware Rewards for Knowledge Reinforcement.** Let $\mathcal{P} = \{p_1, p_2, \ldots, p_{|\mathcal{P}|}\}$ denote the set of predicted bounding boxes and $\mathcal{G} = \{g_1, g_2, \ldots, g_{|\mathcal{G}|}\}$ the ground-truth set. We first establish a one-to-one correspondence between predictions and ground truths using the Hungarian matcher (Carion et al., 2020), with $m = \min(|\mathcal{P}|, |\mathcal{G}|)$. For each matched pair $(p_i, g_i)$, we compute the Intersection-over-Union (IoU) and define a correctness indicator:

$$\delta_i = \mathbf{1}[\text{IoU}(p_i, g_i) > \tau], \tag{1}$$

where $\tau$ is a predefined threshold. Motivated by (Zhan et al., 2025), we define the following reward signals:

**Precision Reward:**

$$R_{\text{precision}} = \frac{1}{|\mathcal{P}|} \sum_{i=1}^{m} \delta_i, \tag{2}$$

**Recall Reward:**

$$R_{\text{recall}} = \frac{1}{|\mathcal{G}|} \sum_{i=1}^{m} \delta_i. \tag{3}$$

### 4.3 Logo-Guided Visual Grounded Reasoning

To mitigate shortcut visual reasoning, we explicitly supervise the model's reasoning process. While SFT effectively injects domain-specific brand knowledge, the model also needs to acquire robust reasoning strategies to generalize to unseen brands. Prior approaches often rely on large-scale distilled CoT datasets to regulate multimodal reasoning, which heavily depends on teacher performance and requires extensive rejection sampling. In contrast, GRPO (Shao et al., 2024) allows the model to refine its reasoning using its own sampled trajectories combined with carefully designed reward functions.

In the context of logo recognition, the key factor is accurately identifying the logo itself. Inspired by GRIT (Fan et al., 2025), we encourage the model to output intermediate reasoning cues, specifically the logo's spatial coordinates, which are supervised using IoU-based rewards. To enhance robustness and reduce potential reward exploitation, we also incorporate an LLM-as-a-judge mechanism to evaluate the model's reasoning process.

**Bounding Box Format Reward.** The model is encouraged to provide explicit reasoning evidence in the form of coordinates. A positive reward is granted whenever the model outputs a valid bounding box in the format $[x_1, y_1, x_2, y_2]$.

**Coordinate Clues Reward.** To prevent the model from providing incorrect coordinate evidence or forgetting to output coordinates during reasoning, we supervise the spatial coordinates during the reinforcement learning stage. Following Eqs. (2) and (3), the main difference is that precise localization is not the primary objective at this stage. The IoU threshold can be adjusted adaptively according to annotation quality to avoid penalizing noisy labels.

**Cognitive Trajectory Reward (CTR).** To evaluate the quality of the model's reasoning, we leverage a large MLLM as an expert assessor. The expert receives the task prompt, the model's output, the ground truth, and the scoring criteria, and then produces a reasoning trajectory along with a score ranging from 1 to 5. To avoid exploitation of the expert's scoring tendencies, the reward is applied only if the final answer is correct, and the contribution of the Cognitive Trajectory Reward is downweighted. For training efficiency, entire batches are processed in parallel, resulting in only a 20% increase in computation time.

**Final Reward:**

$$R = \alpha R_{\text{acc}} + (1 - \alpha)(R_{\text{format}} + R_{\text{bbox format}} + R_{\text{precision}} + R_{\text{recall}} + R_{\text{CTR}}) \tag{4}$$

| Base Model | Training Methods | ID | | OOD | |
|---|---|---|---|---|---|
| | | Acc | F1 | Acc | F1 |
| Qwen2.5-VL-32B | Zero-Shot | 72.27 | 72.13 | 74.38 | 84.00 |
| Doubao-Seed-1.6 | | 68.24 | 67.83 | 81.40 | 88.83 |
| Gemini2.5-Pro | | 67.39 | 67.25 | 89.07 | 93.21 |
| GPT-4.1 | | 67.93 | 67.92 | 90.49 | 94.17 |
| Qwen2.5-VL-3B | Zero-Shot | 55.37 | 54.81 | 67.05 | 75.67 |
| | SFT | 80.48 | 80.46 | 70.07 | 80.52 |
| | GRPO | 78.76 | 76.78 | 73.77 | 80.67 |
| | Logo-VGR | **81.89** | **81.15** | **88.25** | **90.84** |
| | | +3.13 | +4.37 | +14.48 | +10.17 |
| Qwen2.5-VL-7B | Zero-Shot | 67.18 | 67.19 | 81.72 | 84.06 |
| | SFT | 82.46 | 82.45 | 78.68 | 83.68 |
| | GRPO | 82.42 | 82.30 | 84.60 | 88.39 |
| | Logo-VGR | **83.25** | **83.14** | **91.98** | **94.37** |
| | | +0.83 | +0.84 | +7.38 | +5.98 |

Table 1: Performance of various MLLMs on the Open-world Logo Recognition Benchmark. Accuracy and F1 score are reported for the four-class classification task. Our method demonstrates significant improvements over the baseline (GRPO), with particularly strong gains on the OOD dataset, highlighting enhanced generalization to unseen brands.

## 5 EXPERIMENTS

### 5.1 IMPLEMENTATION DETAILS

**Training Details.** All experiments are conducted on eight NVIDIA A800 GPUs. In the Logo Perception Groudning Stage, we adopt LLaMA-Factory(Zheng et al., 2024) as the SFT training framework for the logo detection task. The learning rate is set to $2 \times 10^{-5}$, and training is performed for 2 epochs on 30w logo detection samples. The batch size is fixed at 4, with a dynamic input resolution ranging from $224 \times 224$ to $512 \times 512$. For the GRPO Reinforcement Stage, we employ the Easy-R1(Zheng et al., 2025a) framework. The learning rate is set to $1 \times 10^{-6}$, and gradient updates are accumulated every 32 samples. The KL coefficient is set to $1 \times 10^{-3}$, with each sample drawn 8 times. The IoU reward threshold $\tau$ is fixed at 0.5.

In the second stage, to facilitate the model in predicting detection coordinates across multiple images, we concatenate several product images either horizontally or vertically into a single image. The dynamic resolution is set to range from $224 \times 224$ to $1024 \times 1024$. We again adopt the Easy-R1 framework with a learning rate of $1 \times 10^{-6}$, updating gradients every 32 samples, setting the KL coefficient to $1 \times 10^{-3}$, sampling each instance 8 times, and lowering the IoU reward threshold $\tau$ to 0.3. We set the weight $\alpha$ in Eq. (4) of the Acc reward to 0.5.

**Evaluation Details.** We report results using accuracy (Acc) and F1-score as the main evaluation metrics. For zero-shot methods, since models exhibit varying levels of instruction-following ability, we utilize Doubao-Seed-1.6 to assist in extracting the model's responses.

For detection metrics, we employ pycocotools(COCO Consortium) to evaluate the average precision (AP) of the models at IoU = 0.5. In addition, we use Eqs. (2) and (3) to evaluate the precision and recall of model predictions.

### 5.2 EVALUATION ON OPEN-WORLD LOGO RECOGNITION BENCHMARK

**Performance of Logo-VGR.** We evaluate Logo-VGR on Qwen2.5-VL-3B(Bai et al., 2025) and Qwen2.5-VL-7B. Both models, after being trained with conventional SFT or GRPO, show im-

| Training Methods | LPG | Acc | Format | CTR | Bbox Format | Precision Recall | ID | OOD |
|---|---|---|---|---|---|---|---|---|
| SFT | | | | / | | | 80.48 | 70.77 |
| GRPO | | ✓ | ✓ | | | | 78.76 | 73.77 |
| | | ✓ | ✓ | ✓ | | | 79.87 | 83.50 |
| | | ✓ | ✓ | ✓ | ✓ | | 79.66 | 82.30 |
| | ✓ | ✓ | ✓ | ✓ | ✓ | | 81.26 | 86.03 |
| | ✓ | ✓ | ✓ | ✓ | ✓ | ✓ | 81.89 | 88.25 |

Table 2: Ablation study on the intermediate reward components in Logo Visual Grounding. LPG denotes Logo Perception Grounding, and CTR stands for Cognitive Trajectory Reward. Reported are the accuracies on both the ID and OOD test sets.

provements over the zero-shot baseline on both the ID and OOD test sets. SFT achieves superior performance on the ID set, whereas GRPO demonstrates stronger generalization on the OOD set. Nonetheless, both methods still suffer from performance degradation on OOD. In contrast, Logo-VGR surpasses GRPO by 3 points on the ID set and by 14 points on the OOD set, highlighting its effectiveness in substantially enhancing generalization to unseen brands. As model scale increases, Qwen2.5-VL-7B itself demonstrates substantial improvements in generalization. Remarkably, Logo-VGR outperforms even larger models, such as GPT-4.1.

**Performance of State-of-the-Art MLLMs.** We further assess several state-of-the-art MLLMs, including Qwen2.5-VL-32B(Bai et al., 2025), Doubao-Seed-1.6(ByteDance Seed, 2025), Gemini-2.5-Pro(Comanici et al., 2025), and GPT-4.1(Achiam et al., 2023), on our benchmark. On the ID dataset, these models achieve relatively modest performance, with accuracy around 70%, primarily due to the lack of pre-training in this specific domain. However, their strong generalization ability enables them to perform substantially better on the OOD dataset compared to ID. This improvement can be attributed to the fact that the OOD set consists largely of less-popular brands whose logos exhibit fewer variations and are less affected by counterfeits or adversarial perturbations.

## 5.3 ABLATION STUDIES

We performed ablation experiments on different components of Logo-VGR, including the enhanced perception training in Sec. 4.2 (LPG) and the reward design in Sec. 4.3.

**Effectiveness of Cognitive Trajectory Supervision.** As shown in Table 2, incorporating the Cognitive Trajectory Reward—where the MLLM supervises the model's reasoning process—leads to a substantial performance gain on the OOD dataset. This highlights the crucial role of guiding models to adopt a principled reasoning paradigm.

**Reasoning with Coordinate Clues.** We observe that without Logo Perception Grounding, spatial coordinate outputs fail to enhance reasoning ability, as coordinate generation largely relies on robust logo perception. Once Logo Perception Grounding is introduced, the model's capability improves markedly. Moreover, additional supervision over coordinates reinforces intermediate visual cues and further enhances generalization.

**Analysis of Grounding Metrics.** We evaluate the model's logo grounding capability using AP50, precision, and recall on the Openbrand benchmark(Jin et al., 2020), as illustrated in Tables 3 and 4. Here, AP50 denotes the Average Precision at an IoU threshold of 0.5, which reflects the alignment accuracy between predicted bounding boxes and the ground truth.

As shown in Table 3, the baseline Qwen2.5-VL exhibits limited grounding ability for domain-specific logos, with an AP50 of only 0.061. After SFT training, the model demonstrates substantial improvement in generating accurate logo coordinates, achieving an AP50 of 0.628. With additional Spatial Clues Rewards (precision and recall supervision), the $AP_{50}$ further increases by over 15 points, while precision and recall both improve by nearly 7 percentage points.

You are a brand governance expert of an e-commerce platform. You need to determine which brand the product belongs to based on the product information and the provided product image.
The product image is: <image>
You need to judge which brand the product belongs to.
1.Prada (Reference logo: <image>)
2.Arena (Reference logo: <image>)
3.Tuopin (Reference logo: <image>)
4.None of the above

In the image, the Prada logo can be observed in the regions [333, 224, 418, 311], [919, 224, 999, 311], [1344, 129, 1444, 207], and [1550, 186, 1649, 266].
A closer inspection of the logo details shows that its design characteristics are fully consistent with the official Prada logo reference:
1. Shape: The Prada logo is triangular, featuring three parallel fine lines inside that form a diamond-shaped inner frame.
2. Color: The inner frame of the triangle is white, while the outer edge is black.
3. Edge Treatment: The triangle edges display slight serrated detailing, which perfectly matches the logo details on each shoe shown in the image.
Comparison with reference logos:
1. Prada: Matches all of the above characteristics.
2. Arena: Arena logo consists of two diagonal diamond shapes with two horizontal fine lines in the middle, which does not match any of the Prada logos on the shoes in the image.
3. Tuopin: The Tuopin logo features a white diamond-shaped pattern on a black background, which does not match any of the Prada logo characteristics in the image.
All logo features appearing in the image are entirely consistent with those of the official Prada logo, with no characteristics matching either the Arena or Tuopin logos.

Figure 5: Visualization result of Logo-VGR. The red boxes indicate the predicted coordinates generated by the model. Inference is performed by comparing the logo features extracted from the product image (shown in green) with those of the candidate logos (shown in blue).

Moreover, during the Logo-Guided Visual Grounded Reasoning stage, we investigate the effect of supervising Coordinate Clues (Table 4). Without such supervision, the model tends to exploit the Bounding Box Format Reward by outputting coordinates without meaningful grounding. This highlights the necessity of supervising intermediate coordinates to ensure robust reasoning in downstream tasks.

Table 3: Logo grounding performance comparison on $AP_{50}$, Precision, and Recall.

| Method | $AP_{50}$ | Precision | Recall |
|---|---|---|---|
| Zero-shot | 0.061 | – | – |
| SFT | 0.628 | 81.1 | 82.9 |
| Logo-VGR | 0.780 | 94.2 | 87.4 |

Table 4: Effect of Coordinate Clue Supervision.

| Method | Precision | Recall |
|---|---|---|
| w/o Coordinate Sup. | 0.8 | 0.7 |
| w/ Coordinate Sup. | 61.9 | 58.5 |

## 5.4 QUALITATIVE ANALYSIS

We visualize the model's answers along with the textual coordinate outputs, as illustrated in Fig. 5. The model correctly provides visual coordinates as supporting evidence, compares the detected logo features with those of the candidate reference logos, and then outputs the final decision while clearly explaining the rejection of alternative options. Additional visualization results are provided in the appendix.

## 6 CONCLUSION

In this work, we present Logo-VGR, a novel paradigm for domain-specific multimodal reasoning in intelligent product moderation. By reformulating logo recognition as a comparison-based task, our approach circumvents the impracticality of memorizing massive brand vocabularies and instead focuses on robust reasoning over product–logo pairs. Through the integration of Logo Perception Grounding and Logo-Guided Visual Grounded Reasoning, Logo-VGR effectively mitigates overfitting to brand distributions and significantly improves generalization to unseen brands. Extensive experiments demonstrate that Logo-VGR achieves substantial performance gains over strong baselines, particularly in OOD settings. These findings highlight the potential of domain-specific multimodal reasoning frameworks in advancing real-world applications of MLLMs for product moderation and beyond.

ETHICS STATEMENT

Our dataset is constructed based on publicly available product data from online shopping platforms. We ensured that no personally identifiable information or sensitive user data was collected. The dataset only contains product-level information that is already publicly accessible. Therefore, our work does not involve human subjects, privacy violations, or ethical risks beyond those addressed by using publicly available resources.

REPRODUCIBILITY STATEMENT

To ensure the reproducibility of our research, we will release the full dataset and the code used in our experiments upon publication. The main paper and appendix provide detailed descriptions of the data collection and preprocessing steps, the model architectures, and the training procedures. We believe these resources will enable other researchers to fully reproduce and extend our findings.

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

# A APPENDIX

## A.1 USE OF LARGE LANGUAGE MODELS

In this work, Large Language Models (LLMs) were used as auxiliary tools during the writing and polishing of the manuscript. The details are as follows:

- **Scope of Use**: LLMs were only employed for language polishing, sentence refinement, and improving clarity of expression. They were not used to generate core experimental data, model outputs, or quantitative results. All academic conclusions, experimental setups, and evaluation metrics were independently conducted and validated by the authors.

- **Human Oversight**: All suggestions and outputs from LLMs were carefully reviewed, revised when necessary, and finally confirmed by the authors. The authors take full responsibility for the final content of the manuscript.

- **Ethics and Compliance**: The authors adhered to academic and institutional guidelines in the use of LLMs, ensuring that unverifiable or unauthorized information was not incorporated into the research conclusions.

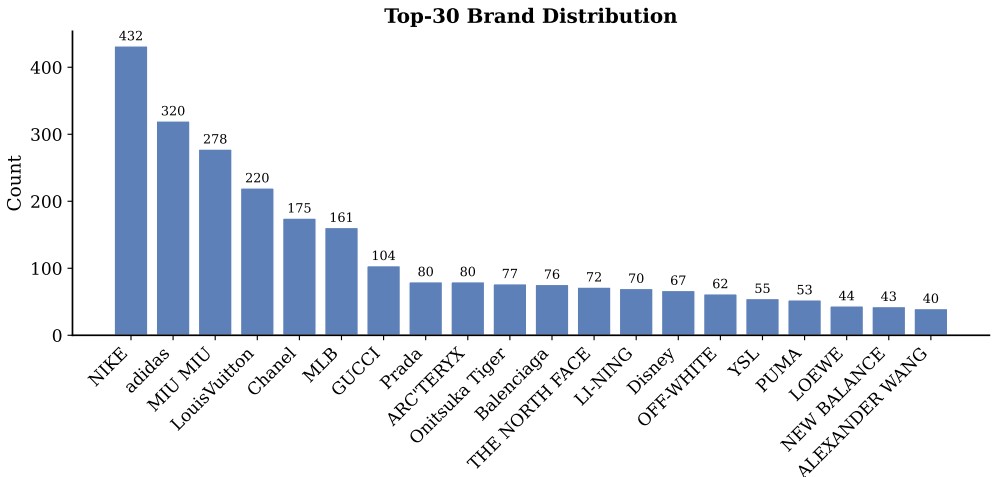

Figure 6: Distribution of the top 30 brands in the training set.

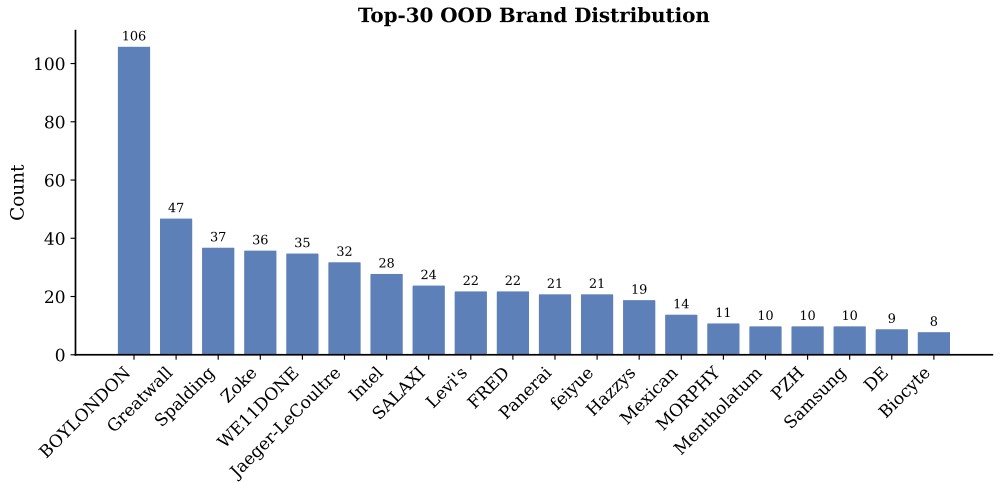

Figure 7: Distribution of OOD brands in the OOD test set.

## A.2 PROMPT FOR COGNITIVE TRAJECTORY EVALUATION WITH LLMS

We provide the detailed prompt used with Doubao-Seed1.6 for evaluating Cognitive Trajectories:

Role: You are a reinforcement learning reward modeling expert, responsible for scoring the quality of the Assistant's responses.

Evaluation Criteria: 1. Does the answer clearly explain the judgment basis? For example, when distinguishing between genuine and counterfeit products, it should point out specific differences between the given logo and the authentic one. 2. Does the answer demonstrate reasoning, with a concise and logical thought process? 3. Is the answer accurate, without hallucinations in the description of images?

Original Task Description: {prompt_str}

Assistant's Response: {response_str}

Ground Truth: {ground_truth}

Please, based on the above criteria, output the scoring rationale and the total score (an integer from 1 to 5). The reasoning should not exceed 100 words.

Output Format: `<think>(Briefly write the scoring rationale)</think> <answer>(Fill in the total score:  an integer between 1 and 5)</answer>`

## A.3 MORE DETAILED DATASET SPECIFICATIONS

We visualize the brand distributions in our dataset, including both the training set and the OOD test set, as shown in Figs. 6 and 7. The brand distributions exhibit a long-tail pattern, with head brands such as "NIKE" and "adidas" occupying the vast majority of the data. Therefore, it is crucial to learn a generalizable reasoning strategy on these head brands, which can then be effectively transferred to unseen OOD brands.

## A.4 ADDITIONAL VISUALIZATION RESULTS

We provide additional visualizations in Figs. 8 and 9. The model generates coordinate clues and describes the logo features in the product images. It then compares these features with each candidate logo and finally produces the prediction.

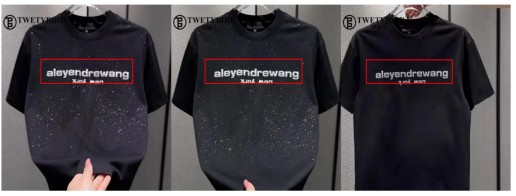

Figure 8: Additional Visualization.

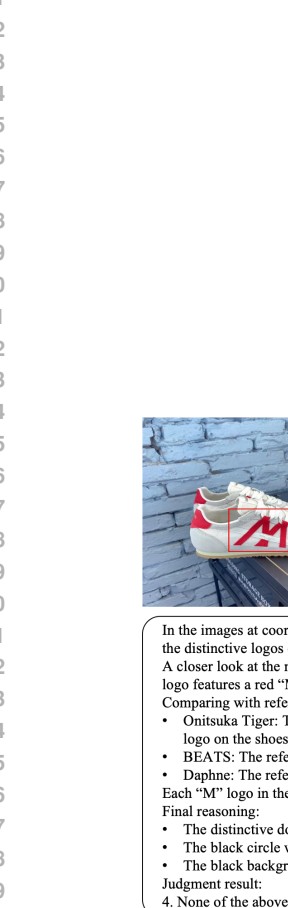
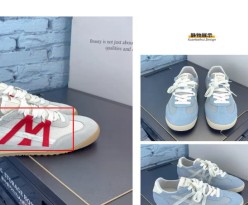
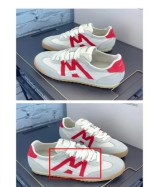

You are a brand governance expert of an e-commerce platform. You need to determine which brand the product belongs to based on the product information and the provided product image.
The product image is: <image>
You need to judge which brand the product belongs to.
1.Onitsuka Tiger(Reference logo: <image>)
2.BEATS(Reference logo: <image>)
3.DaphneG(Reference logo: <image>)
4.None of the above

In the images at coordinates **[256, 323, 501, 476]** and **[1206, 500, 1360, 606]**, multiple uppercase "M" logos can be observed, but none of the distinctive logos of Onitsuka Tiger, BEATS, or Daphne are present.
A closer look at the middle side of the shoes reveals a small "Levi's" logo, indicating that the shoes are branded Levi's. The official Levi's logo features a red "M" with small holes inside the letter and the brand name "Levi's" below it.
Comparing with reference logos:
- Onitsuka Tiger: The reference logo is white letters on a black background with double parallel Tiger stripes, which do not match any logo on the shoes in the images.
- BEATS: The reference logo is a white letter "b" inside a black circle, which does not match any logo on the shoes.
- Daphne: The reference logo is white uppercase letters on a black background, which also does not match any logo in the images.
Each "M" logo in the images perfectly matches Levi's official logo and does not correspond to any other brand's logo features listed above.
Final reasoning:
- The distinctive double parallel Tiger stripes of Onitsuka Tiger are absent.
- The black circle with white "b" of BEATS is absent.
- The black background with white Daphne letters is absent.
Judgment result:
4. None of the above.

Figure 9: Additional Visualization.

