# OpenReview forum: "Logo-VGR: Visual Grounded Reasoning for Open-world Logo Recognition"
_ICLR.cc/2026/Conference — ICLR 2026 Conference Withdrawn Submission_

### Official Review · Reviewer_dXwz · 2025-10-27

**Soundness:** 2
**Presentation:** 2
**Contribution:** 2
**Rating:** 4
**Confidence:** 4

**Summary:**

This paper presents a visual grounding reasoning method for open-world logo recognition. The proposed method adopts a two-stage framework. In the first stage, the model is trained to acquire logo grounding capability using SFT and GRPO, enabling it to predict logo coordinates within product images. In the second stage, the model is trained to select the correct logo for a given product image by generating an intermediate chain of reasoning, guided by several task-specific rewards—such as the Intersection over Union (IoU) reward for the logo’s spatial coordinates and a cognitive trajectory reward evaluated by a LLM. The paper also introduces a logo benchmark comprising 10k QA pairs to evaluate the method, where the test split includes both in-distribution (ID) and out-of-distribution (OOD) questions. Experimental results demonstrate that the proposed reasoning model outperforms zero-shot, SFT, and conventional GRPO baselines on both ID and OOD splits.

**Strengths:**

1.	The paper introduces a visual grounding reasoning framework for logo recognition, which is an interesting and novel research direction.
2.	It develops task-specific rewards to guide the reasoning process, which prove to be effective for the logo recognition task.

**Weaknesses:**

1.	In my understanding, the main difference between the proposed method and conventional GRPO appears to be the use of additional task-specific rewards, which may not constitute a strong novelty.
2.	The details of the benchmark construction are unclear. For instance, the paper does not explain how the data and annotations were collected. Additionally, the advantages of this benchmark over existing ones, such as OpenBrand (Jin et al., 2020), LogoDet-3K (Wang et al., 2022), and SalECI (Jiang et al., 2022), are not explicitly discussed..
3.	In equation 4, only R_{precision} and R_{recall} are defined in the paper. Others are not clearly defined in the paper or appendix.
4.	The experimental comparison is limited to MLLMs with different training strategies. The paper should also compare against conventional logo recognition methods (e.g., Logo-YOLO (Wang et al., 2022)) and recent models such as:
[1] Mark et al., Image-Text Pre-Training for Logo Recognition, WACV.
[2] Dimosthenis et al., Retrieval-Based Methodology for Few-Sample Logo Recognition, 2025

**Questions:**

See weakness.

---

### Official Review · Reviewer_UzmA · 2025-10-29

**Soundness:** 2
**Presentation:** 1
**Contribution:** 2
**Rating:** 2
**Confidence:** 3

**Summary:**

This paper proposes Logo-VGR, a domain-adaptive MLLM framework for open-world logo recognition. Instead of classifying brands directly, it reformulates the task as a comparison problem between product images and candidate logos. The method includes a two-stage training strategy: Logo Perception Grounding for domain knowledge transfer and Logo-Guided Visual Grounded Reasoning for improving generalization via coordinate and reasoning supervision. Experiments on a new benchmark demonstrate strong gains, especially on unseen brands.

**Strengths:**

The problem of logo recognition is interesting and could be practical.

**Weaknesses:**

1. The writing needs to be improved. There are many replicated sentences in the paper. Many statements in the introduction and methodology are identical.

2. Reformulating logo generation into a comparison-based task seems unreasonable to me. Though classification may face generalization problems, recent research (such as CLIP) has proposed to reformulate it as a retrieval problem and has demonstrated impressive zero-shot performance. The paper did not discuss their advantage over retrieval.

3. In experiments, GPT-4’ s zero-shot performance has shown strong enough performance (even stronger than the proposed fine-tuned smaller model), which undermines the necessity of tuning involved in the proposed method. In fact, I doubt using more sophisticated prompting techniques (such as CoT or ICL) can lead to even better performance for GPT-4.

4. Experiments are only conducted on QWen LLMs. More other LLM variants are recommended such as Phi, Gemma, etc.

**Questions:**

See weakness above

---

### Official Review · Reviewer_hB9d · 2025-10-30

**Soundness:** 3
**Presentation:** 3
**Contribution:** 2
**Rating:** 4
**Confidence:** 4

**Summary:**

The paper studies open-world logo recognition for product moderation using multimodal large language models (MLLMs). It proposes Logo-VGR, which reformulates the task as comparison-based matching (product image ↔ candidate logos) instead of direct classification. The method combines Logo Perception Grounding (domain-specific visual grounding) and Logo-Guided Visual Grounded Reasoning (reasoning supervision) to improve generalization. A new benchmark with in-domain (ID) and out-of-domain (OOD) brand splits is introduced, and experiments show around 10-point improvement in OOD accuracy over strong MLLM baselines.

**Strengths:**

The paper’s strengths lie in its creative task formulation and clear empirical gains. Its originality arises from applying reasoning-oriented MLLM training to an open-world logo recognition problem. It provides a newly created dataset with ID/OOD splits allowing clearer measurement of generalization, which could become a useful community resource, to some extends.

**Weaknesses:**

The paper doesn’t compare against existing logo recognition works such as OpenBrand and other traditional approaches, making it hard to judge what’s truly benefit.

The proposed benchmark is also not large-scale, with only about 7,400 training pairs across 373 brands and 109 unseen brands for testing, which limits how convincingly it represents the “open world” the paper aims to model.

In addition, the method assumes that suitable candidate logos are already available for comparison, but it does not discuss how these reference logos would be collected, updated, or retrieved efficiently from large databases. Likewise, it remains unclear how the proposed model would integrate into or improve existing moderation systems beyond the benchmark setting.

**Questions:**

It looks to me the proposed framework will be a great incremental enhancement to existing logo detection or retrieval systems. It could serve as a booster in the ranking stage if we think about the logo recognition as a two stage problem: retrieval and ranking. I'm looking for more details to describe how candidate logos could be sourced, how the model scales to massive logo inventories, and how it would concretely boost current system performance—would make the work much more compelling and impactful.

---

### Official Review · Reviewer_bz5A · 2025-10-30

**Soundness:** 3
**Presentation:** 2
**Contribution:** 2
**Rating:** 4
**Confidence:** 4

**Summary:**

This paper introduces Logo-VGR, a two-stage training paradigm designed to tackle the open-world logo recognition problem. The authors reframe the problem from a memory-intensive classification task to a comparison-based reasoning task, which is more scalable and generalizable.  The authors demonstrate through extensive experiments on their custom-built benchmark that Logo-VGR significantly outperforms strong baselines, especially on out-of-domain (OOD) brands.

**Strengths:**

1. The paper's core insight is reasonable, i.e. to supervise the entire reasoning process, using coordinate and text-quality rewards to ensure the model understands the task rather than just memorizing answers.

2. This method yields compelling results, e..g, in accuracy on unseen brands.

3. The ablation studies show the effectiveness of each component.

**Weaknesses:**

Major:
1. A primary concern revolves around the newly introduced benchmark and its ID vs OOD split. The paper states that head brands form the training set and tail brands form the OOD set, which mimics a real-world long-tail distribution. However, this could introduce confounding variables. Are the OOD brands inherently easier to recognize (e.g., having simpler or more distinct visual features)? The paper notes that large models perform better on OOD data because these brands have "fewer variations and are less affected by counterfeits," which suggests a potential disparity in difficulty that is not just about being "unseen." A more detailed analysis of the visual complexity and intra-class variation of ID vs. OOD brands is needed.

2. I think it would be beneficial to validate the Logo-VGR method on established public logo recognition datasets.

3. The model relies on an "coarse retrieval model" that provides candidate brands. The performance of this module directly sets the upper bound for the main task and modulates its difficulty.  It should report the recall@3 of this retrieval model and discuss how frequently the "None of the above" option is the correct answer.

4. The paper does not discuss the framework's dependency on the specific LLM chosen for LLM-as-a-judge.

Minor:

1.  In Line 357, the use of "30w" is informal for a paper. It should be written as "300,000" or "300K".

**Questions:**

Please see the Weaknesses. I would like to raise the score if the concerns have been properly addressed.

---

### Note · Authors · 2025-11-14

I have read and agree with the venue's withdrawal policy on behalf of myself and my co-authors.